# Essential Oil-Based Soap with Clove and Oregano: A Promising Antifungal and Antibacterial Alternative against Multidrug-Resistant Microorganisms

**DOI:** 10.3390/molecules29194682

**Published:** 2024-10-02

**Authors:** Ana Paula Merino Cruz, Felipe Garcia Nishimura, Vinícius Cristian Oti dos Santos, Eliana Guedes Steling, Marcia Regina Von Zeska Kress, Mozart Marins, Ana Lucia Fachin

**Affiliations:** 1Unidade de Biotecnologia, Universidade de Ribeirão Preto (UNAERP), Ribeirao Preto 14096-900, Brazil; ana.pcruz@sou.unaerp.edu.br (A.P.M.C.); felipegnishi@hotmail.com (F.G.N.); vinicius.csantos@sou.unaerp.edu.br (V.C.O.d.S.); mmarins@unaerp.br (M.M.); 2School of Pharmaceutical Sciences of Ribeirão Preto, University of São Paulo (USP), Ribeirão Preto 14040–903, Brazil; elianags@usp.br (E.G.S.); kress@fcfrp.usp.br (M.R.V.Z.K.)

**Keywords:** essential oils, hand hygiene, antifungal activity, antibacterial activity, antimicrobial soap

## Abstract

The transmission of microorganisms via hands is a critical factor in healthcare-associated infections (HAIs), underscoring the importance of rigorous hand hygiene. The rise of antimicrobial-resistant microorganisms, driven in part by the overuse of antibiotics in clinical medicine, presents a significant global health challenge. Antimicrobial soaps, although commonly used, may exacerbate bacterial resistance and disrupt skin microbiota, posing additional health risks and environmental hazards. Essential oils, with their broad-spectrum antimicrobial properties, offer a promising alternative. This study evaluates the antimicrobial activity of essential oils against various bacterial and fungal strains, including multidrug-resistant isolates. Using a range of in vitro and in vivo antimicrobial assays, including minimal inhibitory concentration (MIC), minimal bactericidal concentration (MBC), and minimal fungicidal concentration (MFC), the essential oils were tested against a broad spectrum of pathogens. Additionally, the chemical composition of the oils was analyzed in detail using gas chromatography–mass spectrometry (CG–MS). Clove, oregano, and thyme oils demonstrated potent inhibition of all tested ATCC bacterial strains, with MIC values ranging from 3.125 to 50 μL/mL. These oils also showed significant activity against multidrug-resistant *Escherichia coli* and *Pseudomonas aeruginosa* strains. Notably, clove oil exhibited remarkable efficacy against fungal strains such as *Aspergillus fumigatus* and *Trichophyton rubrum*, with MIC values as low as 1.56 μL/mL. Synergy tests revealed that combinations of clove, oregano, and thyme oils yielded significantly lower MIC values than individual oils, indicating additive or synergistic effects. The formulation of a soap incorporating clove and oregano oils demonstrated efficacy comparable to synthetic antiseptics in vivo. These findings highlight the exceptional antimicrobial potential of essential oils, mainly clove and oregano, against resistant microorganisms, offering a viable alternative to conventional antimicrobial agents.

## 1. Introduction

The integumentary system, comprising the skin, represents the human body’s largest organ and fulfills multifaceted roles crucial for sustaining life. Supplied with distinct anatomical features by its strategic localization, the skin accommodates a diverse array of microorganisms, delineating a dynamic ecosystem [1].

The hands, being the principal conduits for dissemination of these microorganisms, assume a pivotal role in the transmission dynamics. Within healthcare domains, the hands of healthcare practitioners constitute the primary conduit for microbial transfer, thereby underscoring the importance of stringent hand hygiene practices for preventing healthcare-associated infections (HAIs) [2].

HAIs engender deleterious ramifications post-hospitalization, potentially manifesting during convalescence or post-discharge. The repercussions of HAIs are manifold, encompassing escalated morbidity and mortality rates, augmented financial burdens, and a compromised landscape for patient safety, thereby undermining the quality of healthcare delivery [3].

Numerous studies have elucidated the alarming rise of microorganisms exhibiting resistance to diverse antimicrobial agents, constituting a pressing global health concern [4]. This surge in antimicrobial resistance stems from the widespread and often excessive utilization of antibiotics, prevalent not only within clinical settings but also in animal feed and agricultural practices. Consequently, a surge in selective pressure ensues, creating opportunities for microorganisms to encounter antimicrobial agents, thereby catalyzing the development of resistance through the acquisition of resistance genes or mutations in determinants of resistance [4,5,6].

Antimicrobial soaps, laden with potent antiseptic agents, potentially exacerbate bacterial resistance through repeated and indiscriminate usage. Moreover, these agents can disrupt the equilibrium of the skin’s normal microbiota, thereby predisposing individuals to opportunistic infections. Furthermore, certain antiseptic soaps harbour carcinogenic compounds, eliciting adverse reactions such as allergies and endocrine disruptions [7,8]. Moreover, domestic and industrial disposal of these products poses grave environmental risks, culminating in bioaccumulation and persistent threats to aquatic organisms and human health [9].

Chlorhexidine and triclosan, ubiquitous antiseptic agents used in hand hygiene, have been implicated in contact dermatitis cases. Recent studies suggest an underestimation of the allergic potential of chlorhexidine, warranting scrutiny, particularly among healthcare professionals, given their frequent exposure [10]. Additionally, the FDA’s prohibition of triclosan and triclocarban in 2016 underscores the recognition of the adverse effects of these agents in handwashing products [11].

Exploration of alternative antiseptics is imperative, given the growing reports on microbial adaptability to conventional agents. Essential oils, derived from diverse botanical sources, present a promising alternative in combating resistant microorganisms owing to their multifaceted antimicrobial properties [12,13]. Thus, concerted efforts are warranted to harness the therapeutic potential of natural compounds in mitigating the effects of antimicrobial resistance.

Essential oils exhibit a rich repertoire of biological activities, prominently featuring antibacterial, antifungal, insecticidal, and antiviral properties [14,15]. Attributed to their chemical constituents, particularly terpenes and phenylpropanoids, essential oils are renowned for their distinctive fragrances and potent therapeutic effects. The presence of terpenes, phenolics, and aldehydes charge these oils with profound biomedical significance, offering a formidable arsenal against a spectrum of pathogenic invaders, including viruses, fungi, and bacteria. Moreover, essential oils may harbour an array of complementary compounds such as fatty acids, oxides, and sulfur derivatives, augmenting their multifaceted utility [16].

Amongst the plethora of essential oils, those derived from clove, oregano, rosemary, and thyme stand out for their efficacy in curbing the proliferation of microbial strains that have developed resistance to conventional antibiotics. Noteworthy is the compelling efficacy of clove oil against the methicillin-resistant Staphylococcus aureus strain, underscoring its potential as a formidable antimicrobial agent [17].

In light of these insights, the primary objective of this study was to evaluate the antimicrobial activity of essential oils and explore their potential as constituents in the formulation of novel antiseptic agents.

## 2. Results

### 2.1. Antibacterial Activity

The essential oils of clove, oregano, and thyme inhibited all tested ATCC bacterial strains with MIC values ranging from 1.56 μL/mL to 50 μL/mL (Table 1). It is noteworthy that the oils also exhibited promising results against multidrug-resistant strains *E. coli* EW222 and *E. coli* EW239, with MIC values ranging from 3.12 to 25 μL/mL for these strains. Furthermore, the same oils also showed activity against *P. aeruginosa* S15, isolated from soil samples, with MIC values of 12.5 μL/mL for clove and oregano, and 50 μL/mL for thyme.

Ethanol was utilized as a solvent to dilute the oils, with its final concentration not exceeding 1% in the highest treatment. In all inhibition assays, ethanol was included as a control and demonstrated no inhibitory activity.

Given these results, which were superior to those found with clove, oregano, and thyme oils when comparing MIC values against these same tested bacterial strains, we decided not to perform MIC tests with the previously tested bacterial strains. Additionally, 10% ethanol did not inhibit any of the tested strains, indicating that its presence in the oil dilution did not interfere with the results.

Regarding the inhibitory effect evaluated by the MBC, most oils exhibited bactericidal effect (Table 2). Bacteriostatic activity was observed in clove oils for *P. aeruginosa* S15, clove, oregano, and thyme for *S. epidermidis* ATCC 12228, eucalyptus oil for *S. aureus* ATCC 6538, and clove, thyme, and oregano oils for *Salmonella Choleraesuis* ATCC 10708.

### 2.2. Antifungal Activity

The most notable results against the tested filamentous fungal strains were observed with oregano, clove, and thyme oils. Particularly noteworthy is clove oil, which inhibited the *A. fumigatus* LMC 9015.01 strain (isolated from patient) and *T. rubrum* (ATCC MYA 3108) with an MIC value of 1.56 μL/mL (Table 3). The solvent control conducted with ethanol did not inhibit the growth of any of the fungal strains. Furthermore, all tested essential oils of oregano, clove, and thyme exhibited fungistatic action.

Regarding yeast strains, the best MIC results were also obtained with oregano, clove, and thyme oils (Table 4). It is worth noting that the MIC values of oregano, clove, and thyme oils for *C. auris* CDC 811903 were lower than that of fluconazole (3.9 μg/mL). The results of the essential oils’ activity against the *C. neoformans* strain were promising, with clove oil being the most effective with an MIC of 6.25 μg/mL, while others ranged from 12.5 to 25 μg/mL. For *C. albicans*, oregano oil exhibited the best growth inhibition with an MIC of 3.12 μg/mL. Regarding the minimum fungicidal concentration, unlike the results for filamentous fungi, several essential oils showed fungicidal action. Oregano and thyme oils stood out for their fungicidal action against the *C. neoformans* ATCC 90112 and *C. albicans* ATCC 1023 strains. Clove oil showed fungistatic activity against *C. neoformans* ATCC 90112 and *C. auris* CDC 811903 and showed fungicidal activity for *C. albicans* ATCC 10231. 

### 2.3. Antimicrobial Activity and Synergism of the Oils

The MIC values of combinations of clove, oregano, and thyme oils were significantly lower when compared to the values obtained from the oils tested individually, indicating additive or synergistic effects among the oils (Table 5). The highest MIC value was observed for the clove/thyme combination at 6.25 μL/mL, which was still half the MIC value obtained with the oils tested individually against *S. epidermidis*. The MIC value for the clove and oregano combination was the lowest among the three combinations.

Regarding the MBC, the clove/oregano combination exhibited bactericidal effects against four out of five tested strains, with bacteriostatic effects observed only against *S. aureus* (Table 6). The thyme/oregano combination showed bactericidal activity against three out of five tested strains, and the clove/thyme combination exhibited bactericidal activity against only two strains.

Additionally, Table 7 shows that the evaluation of the Fractional Inhibitory Concentration Index yielded positive results, except for the *E. coli* EW239 strain (indifferent) in the thyme/oregano combination.

### 2.4. GC-MS

Chromatographic evaluation was performed only on clove, oregano, and thyme oils, as these demonstrated the most potent antibacterial and antifungal activities. GC–MS results revealed that each oil predominantly contains specific molecules likely responsible for the antimicrobial effects observed in inhibitory activity assays. Clove oil was found to contain 84.36% eugenol, oregano oil 72.63% carvacrol, and thyme oil 49% thymol (Table 8).

### 2.5. Evaluation of In Vivo Antimicrobial Activity

The soap containing essential oils exhibited the highest logarithmic reduction factor value (4.55), whereas the other analyzed products showed lower values (3.20 for commercial soap and 3.65 for chlorhexidine) (Figure 1).

Statistical analysis revealed a significant difference between the soap containing essential oils and the base, as well as compared to commercial soap 1, thus confirming the superior performance of the essential oil formulation. Comparison with the base demonstrated that the difference in results occurred due to the addition of clove and oregano essential oils. When compared to chlorhexidine 2%, there was no statistical difference, likely due to the high standard deviation observed in the chlorhexidine experiment.

## 3. Discussion

This study focused on exploring the potential of essential oils in combating various strains of microorganisms, including those resistant to conventional treatments, as well as formulating an antiseptic agent utilizing these oils.

Essential oils are complex mixtures of natural compounds, typically containing 20 to 60 constituents in varying proportions. Among these constituents, carvacrol, thymol, and eugenol (Figure 2) stand out, comprising a significant proportion of oregano, thyme, and clove oils, respectively [5,18,19,20]

The results of GC–MS assays revealed the presence of these molecules in significant concentrations in oregano, thyme, and clove oils, with eugenol in clove oil being the most prominent, reaching a concentration of 84.36%. It is noteworthy that the predominance of these compounds is directly associated with the antimicrobial activity of essential oils [21,22].

Essential oils exert their antimicrobial action by destabilizing the structure of bacterial cell membranes, resulting in their integrity breakdown and increased permeability [18,22] thereby interfering with various vital cellular activities such as energy production and membrane transport. Due to their lipophilic nature, essential oils can easily penetrate bacterial cells. 

Oregano, thyme, and clove oils have demonstrated significant antimicrobial activity, even against multidrug-resistant strains of *E. coli*, with relatively low Minimum Inhibitory Concentration (MIC) values (3.12 to 25 μL/mL). Although the antimicrobial effectiveness of these oils is directly related to the number of active compounds they contain, oregano oil was the most effective, possessing a relatively high concentration of carvacrol (72.63%).

The antimicrobial activity of carvacrol and eugenol have been previously demonstrated. Arfa et al. (2006) emphasized the importance of the hydrophobicity and chemical structure of aromatic phenolic compounds for this activity [23]. Additionally, Vale et al. (2021) observed a synergistic activity when eugenol, carvacrol, and thymol were combined, especially against *Amblyomma sculptum* larvae, known as the star tick, an important vector of the bacterium *Rickettsia rickettsii* and other parasites [24].

Recently, Olaimat et al. (2024) demonstrated that both eugenol and carvacrol exhibit promising activity against pathogenic bacteria causing foodborne infections, such as *S. enterica* and *E. coli* O157:H7 [25]. Furthermore, the combination of thymol and carvacrol also showed synergistic activity against these and other pathogenic species [26].

In addition to bacteria, fungi also represent significant pathogenic agents. Although they constitute a small portion of the human microbiota and many fungi are familiar to the immune system, several can still trigger infections in immunocompromised hosts, and are classified as opportunistic pathogens [27]. Genera such as *Aspergillus, Cryptococcus, Candida,* and *Trichophyton* are responsible for over 75% of global related fatalities from invasive and superficial fungal infections [28,29]. Faced with this scenario, pursuing new therapeutic approaches becomes crucial, considering that conventional antifungals have shown limited efficacy due to the development of resistance by fungi.

The essential oils tested in this study demonstrated promising antifungal activity. Once again, oregano, thyme, and clove oils stood out, exhibiting even lower MIC values than those observed against bacteria. While for most bacteria the oils demonstrated bactericidal activity, for fungi, the activity was predominantly fungistatic against A. fumigatus and *T. rubrum*, and fungicidal for most *Candida* strains, except for *C. auris*, where the oils exhibited only fungistatic activity.

Numerous studies demonstrated the activities of essential oils against fungi. For example, Schroder et al. (2017) investigated the antifungal activity of essential oils against environmental fungi and observed that 0.63% clove oil was able to inhibit the growth of Aspergillus spp. [30]. Kumari et al. (2017) evaluated the antifungal activity of active compounds present in essential oils of oregano (carvacrol), peppermint (menthol), and clove (eugenol) against *C. neoformans* strains, with Minimum Inhibitory Concentration (MIC) ranging from 32 to 256 μg/mL [31]. Parker et al. (2022) also obtained promising results for peppermint, clove, and tea tree oils against *C. auris* strains, highlighting the potential activity of these oils against this fungus, which represents a significant public health challenge due to its rapid ability to develop resistance to multiple drugs [32].

Other studies have explored combinations of essential oils to determine their enhanced antimicrobial activity. Rapper et al. (2023) investigated four combinations of essential oils against nine respiratory tract-associated pathogens, demonstrating superior antimicrobial activity compared to individual essential oils [33]. The combination of clove and oregano oils stood out, especially against the tested strains. Vasconcelos et al. (2020) examined the antibacterial activity of cinnamomum cassia essential oil, both alone and in combination with antibiotics, against *Klebsiella pneumoniae* and *S. marcescens*, highlighting a synergistic effect against *S. marcescens* [34].

Due to the promising results against both bacteria and fungi, oregano, thyme, and clove oils were selected for synergy testing. Except for the combination of thyme and oregano oils, which showed synergistic effects only against the *E. coli* EW239 strain, the other combinations (clove/oregano and clove/thyme) demonstrated additive or synergistic effects against the remaining tested strains.

In addition to resistance developed by microorganisms to antimicrobial agents, resistance to antiseptics can also emerge. Examples include chlorhexidine and commercial soap 1, often used for hand antiseptics by healthcare professionals and the general population, respectively.

Chlorhexidine is a synthetic antiseptic, to which some microorganisms have already demonstrated resistance [35,36,37,38]. Studies indicate that synthetic products can promote microbial resistance and cause adverse reactions, whereas natural products are less likely to induce resistance [7,8]. The formulation of commercial soap 1 is based on natural compounds, contrasting with synthetic products, which follows a market trend. It contains four ingredients with antimicrobial properties: limonene, linalool, *Camellia sinensis* extract, and flaxseed oil.

Limonene, found in essential oils of citrus fruits, exhibits antibacterial, antifungal, and antiviral activity, and also impedes biofilm formation [39,40]. Linalool, present in essential oils of various plants, is recognized for its antioxidant, anti-inflammatory, anticancer, and antibacterial properties [41,42]. Flaxseed oil, rich in phenolic compounds, phenolic acid, flavonoids, lignans, and tannins, exhibits antibacterial and antifungal activity [43].

In the present work, the formulation of the soap containing clove and oregano oils proved to be more effective than commercial soap, although it did not show a statistically significant difference compared to 2% chlorhexidine. However, it is important to note that the formulation based on natural active ingredients performed comparably to chlorhexidine, which uses synthetic actives. As mentioned above, synthetic products have a higher probability of generating resistance by microorganisms.

## 4. Materials and Methods

### 4.1. Essential Oils

The essential oils examined in this study comprised: oil of clove (*Eugenia caryophyllata*) from Oficina de Ervas (Herbal Pharmacy, Ribeirão Preto, SP, Brazil) and oils of oreg4no (*Origanum vulgare*) and white thyme (*Thymus vulgaris*) from Ferquima (Ind. E Com. Ltd.a, Vargem Grande Paulista, SP, Brazil).

For the experiments, the essential oils were diluted in 10% ethanol. For bacterial activity, this solution was further diluted in Mueller Hinton broth (HIMEDIA^®^, Kelton, PA, USA) at a ratio of 900 μL of medium to 100 μL of the 10% ethanol–oil mixture. For antifungal activity, the oil solution was further diluted in RPMI medium (Sigma-Aldrich, St. Louis, MO, USA) in the same ratio.

### 4.2. Bacterial Strains 

Bacterial strains from the ATCC collection, as well as multidrug-resistant bacteria obtained from public aquatic environments and soil, were employed (Table 9). The *P. aeruginosa* S15, *E. coli* EW222, and *E. coli* EW239 are multidrug-resistant strains of *P. aeruginosa* and *E. coli* isolated, respectively, from soil and aquatic environments [44,45]. Bacteria were preserved in Brain Heart Infusion (BHI) medium (HIMEDIA^®^, Kelton, PA, USA) supplemented with 15% glycerol (Sigma-Aldrich, St. Louis, MO, USA) at −80 °C and subsequently cultured from the stock on Mueller Hinton agar (KASVI, São José dos Pinhais, PR, Brazil), and incubated for 24 h at 37 °C, as per the protocol described by Cruz et al. (2019) [46].

### 4.3. Fungal Strains 

Furthermore, filamentous fungi were employed, including *Trichophyton rubrum* ATCC MYA-3108, *Aspergillus fumigatus* ATCC 46645, and a clinical isolate from patient sputum, *Aspergillus fumigatus* LMC 9015.01 [47]. For the assays, fungi were cultured on Sabouraud agar (KASVI, São José dos Pinhais, PR, Brazil) for 7 days at 30 °C.

Subsequently, the essential oils were also tested against yeasts, including *Candida albicans* ATCC 10231, *Candida auris* CDC 811903, and *Cryptococcus neoformans* ATCC 90112. These were cultured in RPMI-1640 medium (Sigma-Aldrich, St. Louis, MO, USA) and buffered with 3–[N-morpholino] propane sulfonic acid (MOPS–SERVA) to a concentration of 0.165 mol/L, for 24 h at 35 °C.

### 4.4. Minimal Inhibitory Concentration

The antimicrobial activity, determined by the minimal inhibitory concentration (MIC) assay of the essential oils, was assessed using the microdilution method in 96 well plates, as described by CLSI M7-A11 (2018) for bacteria, CLSI M38-A3 (2017) for filamentous fungi, and CLSI M27-A4 (2017) for yeasts [48,49,50].

The bacterial inoculum concentration was adjusted using a spectrophotometer at a wavelength of 550 nm to reach absorbance value between 0.100 and 0.125, and then the standard inoculum was diluted 1/50 in Mueller Hinton Broth. The fungal inoculum concentration was adjusted using a spectrophotometer at a wavelength of 530 nm to reach absorbance value between 01.25 and 0.155, and then the standard inoculum was diluted 1/50 in RPMI medium for filamentous fungi, while, for yeasts, another dilution of 1/20 was performed after the 1/50 dilution. Sterility controls of the culture medium and oils were performed concurrently, as well as growth controls. The concentration of the tested oils ranged from 50 µL/mL to 0.000023 µL/mL.

The MIC was defined as the lowest concentration of the oil at which no macroscopic growth of microorganisms was observed compared to the growth of controls. MIC was determined after 24 h of incubation at 37 °C for bacteria, 7 days at 30 °C for filamentous fungi, and after 24 h at 35 °C for yeasts. Gentamicin sulfate 100 µg/mL (Sigma-Aldrich, St. Louis, MO, USA) was used as a positive control for bacteria, fluconazole for filamentous fungi and yeasts, and 8% chlorhexidine digluconate was tested for both. MIC assays were performed in triplicate.

### 4.5. Minimal Bactericidal Concentration and Minimal Fungicidal Concentration

Following the determination of the MIC for different microorganisms, assessments of the minimal bactericidal concentration (MBC) and minimal fungicidal concentration (MFC) of the analyzed oils were conducted, also distinguishing the MIC value from being bactericidal or bacteriostatic and fungicidal or fungistatic.

Using a bacteriological loop, aliquots of the culture medium from wells defined as MIC, as well as from wells of the two previous concentrations and control wells, were inoculated onto the surface of plates containing Mueller Hinton agar for bacteria and Sabouraud agar for fungi and yeasts. These plates were then incubated in a 37 °C incubator for 24 h for bacteria, 30 °C for 7 days for filamentous fungi, and 35 °C for 24 h for yeasts. MBC and MFC were defined as the lowest concentration of the tested oil that resulted in microbial death.

### 4.6. Evaluation of Synergistic Effect of Essential Oils for In Vitro Antimicrobial Activity

To assess the synergistic effect, clove, oregano, and white thyme oils were employed, forming the following combinations: clove/oregano, clove/thyme, and thyme/oregano. After mixing the two oils of each combination, initial concentrations were adjusted to 50 µL/mL.

The synergistic effect of essential oils was evaluated against Gram-positive and Gram-negative bacterial ATCC strains: *E. coli* ATCC 25922, *S. aureus* ATCC 6538, *S. epidermidis* ATCC 12228, *S. marcescens* ATCC 13880, in addition to a bacterial strain isolated from a public aquatic environment, *E. coli* EW239 (multiresistant).

The determination of MIC and MBC followed the same protocol described for the individual assessment of oils against bacteria. The concentration range of the tested oils varied from 25 µL/mL to 0.00004668 µL/mL. The antibiotic gentamicin sulfate (SIGMA^®^) at a concentration of 100 µg/mL and 8% chlorhexidine digluconate were used as positive controls. MIC was determined in two independent experiments, each performed in triplicate.

The combinatory effect of essential oils was evaluated using the “checkerboard” test with culture broth in microtiter plates. The Fractional Inhibitory Concentration Index (FICI) of the most effective oils, which is the sum of the Fractional Inhibitory Concentration (FIC), was evaluated according to the formula (1) described by Karpanen et al. [51].
A/MICa + B/MICb = FIC A + FIC B = FICI(1)
where: 

A = MIC of drug A in combination

MIC a = MIC of drug A alone

B = MIC of drug B in combination

MIC b = MIC of drug B alone

Interaction is defined as follows:

–Synergistic when FICI is less than or equal to 0.50;

–Additive when FICI > 0.50 and less than or equal to 1.0;

–Indifferent if FICI > 1 and less than or equal to 4.0;

–Antagonistic when FICI > 4.0.

### 4.7. Soap Formulation

For the development of the liquid soap formulation, essential oils exhibiting the most potent antimicrobial activities against the tested bacteria, filamentous fungi, and yeasts, based on the MIC tests and their demonstrated synergism, were selected and incorporated into the liquid commercial soap base (10% sodium laureth sulfate, 10% cocamidopropyl betaine, EDTA 0.1%). Due to the strong odour of thyme oil, it was deemed unsuitable for incorporation into the soap formulation. The combination of clove and oregano oils, each at a concentration of 50 µL/mL, was utilized in the soap formulation. This concentration is double the highest MIC value observed during the assays, which was 25 µL/mL against *Pseudomonas aeruginosa* ATCC 27853 for both clove oil and oregano oil. This ensures that the oils would inhibit the majority of strains. 

### 4.8. In Vivo Assay

#### 4.8.1. Preparation of Bacterial Culture for Artificial Contamination of Hands

The bacterial strain used for finger contamination culture was *S. marcescens* ATCC 13880, selected for its reddish pigmentation of colonies, as it facilitates colony count visualization compared to bacteria from normal microbiota [52,53]. It was stored in BHI medium with 15% glycerol at −80 °C and then seeded from the stock onto solid Tryptic Soy Broth (TSB–Sigma–Aldrich, St. Louis, MO, USA) culture medium supplemented with agar and incubated for 24 h at 37 °C. After growth, a colony was isolated in 50 mL centrifuge tubes with 20 mL of TSB, and the culture was then incubated for 24 h at 37 °C. A new growth cycle was initiated by transferring 500 µL of the bacterial solution into 30 mL of TSB medium. Forty-five centrifuge tubes containing 30 mL of TSB medium were prepared and consolidated in a 2 L Erlenmeyer flask. The final bacterial inoculum concentration was adjusted using a spectrophotometer at a wavelength of 620 nm, within an absorbance range of 0.150 and 0.460 (BS EN 1500:2013). In total, 4 liters of bacterial culture were prepared for the hand contamination of 20 volunteers. The sample of 20 healthy individuals met the following inclusion criteria: no contact with chlorhexidine on the day of collection, as it has a residual effect; absence of signs of dryness or skin lesions; absence of signs of fungal infections or skin infections; age between 18 and 55 years, as there are alterations in skin microbiota after the age of 60; and overall good health. These volunteers were divided into 4 groups: tests with the base (soap formulation without essential oils), tests with the soap with essential oils, tests with 2% chlorhexidine digluconate, and tests with the commercial liquid soap. The antimicrobial activity assay of the liquid soap was conducted according to the method used to evaluate the efficacy of hand hygiene agents of EN 1500 [54]. Initially, the volunteers washed their hands with 1 mL of liquid soap without antiseptic activity for 60 s (following the protocol techniques) and then dried them for 30 s with paper towels. Subsequently, they partially immersed their fingers (up to half of the metacarpals) into the bacterial culture broth for 5 s. The hands were then removed from the culture broth, allowing the excess fluid to drain by tilting them downwards, and were then air-dried for 3 min in a horizontal position.

Immediately after drying, to obtain the colony count (pre-values), the fingers of each hand were separately rubbed onto two Petri dishes for 60 s in 10 mL of Tryptic Soy Broth (TSB) medium (SIGMA-ALDRICH^®^).

Following this, 1 mL of the test soap (containing essential oils) was applied for hand washing, which was performed within a maximum time of 60 s, also following the protocol techniques. To obtain the colony-forming unit count after soap use (post-values), the hands were dried for 30 s, but now in the open air without using paper towels, and the fingers were rubbed onto Petri dishes with 10 mL of TSB medium again for 60 s. The same procedure was carried out with the soap base (without the presence of oils), with the reference product 2% chlorhexidine digluconate, and with a second commercial liquid soap. Next, dilutions of the pre-values broth (after finger contamination) were made from 10^−1^ to 10^−5^ to facilitate colony counting. Using a pipette, 100 µL of each dilution of 10^−3^, 10^−4^, and 10^−5^ were seeded onto the surface of Petri dishes with TSA medium. The plates were then incubated for 24 h at 37 °C. The same procedure was performed for the post-values broth (after hand washing), but the dilutions were adjusted to 100, 10^−1^, and 10^−2^.

#### 4.8.2. Analysis of In Vivo Assay Results

Colony counting was performed after 24 h of incubation at 37 °C. The colony-forming units (CFU) count of the right and left hands of the 5 volunteers was determined for both pre-values and post-values. Subsequently, the logarithm of the weighted average of the last two dilutions was calculated, namely, 10^−4^ and 10^−5^ (for pre-values) and 10^−1^ and 10^−2^ (for post-values).
Log 10 of Z = ƩC/v1 × d1 + v2 × d2(2)
where:

Z = weighted average of CFU/mL of the pre- or post-value sampling fluid;

ΣC = sum of CFUs counted on the plates used for calculation;

v1 = volume of the inoculum in mL, taken from the sampling fluid at the lowest dilution;

v2 = volume of the inoculum in mL, taken from the sampling fluid at the highest dilution;

d1 = dilution factor corresponding to the lowest dilution of the sampling fluid;

d2 = dilution factor corresponding to the highest dilution of the sampling fluid.

The logarithmic reduction factor was calculated, which is the difference between the logarithm of the mean pre-values (considering the sum of the right and left hands) and the logarithm of the mean post-values (considering the sum of the right and left hands), for each of the hand hygiene procedures performed by the volunteers. 

For statistical analysis, one-way ANOVA followed by Bonferroni’s multiple comparisons test was applied to the logarithmic reduction factor values of each volunteer for each tested product.

### 4.9. Analysis of Essential Oil Constituents by Gas Chromatography–Mass Spectrometry (GC–MS)

Chemical constituents present in the essential oils were analyzed using GC–MS (gas chromatography–mass spectrometry) with a Varian 3900 instrument equipped with a Saturn 2100T mass-selective detector. The analysis conditions were as follows: capillary column: DB-5 (30 m × 0.25 mm × 0.25 μm); injector temperature: 240 °C; detector temperature: 230 °C; electron impact: 70 eV; carrier gas: helium; flow rate: 1.0 mL/min; split ratio: 1/20; temperature program: 60 °C–240 °C, 3 °C/min; injection volume: 1 μL of solution (1 μL essential oil/1 mL AcOEt). 

Substance identification was conducted by comparing their mass spectra with the database of the GC–MS system (NIST 62 lib.) and the Kovats retention index.

## 5. Conclusions

Microbial infections pose a significant threat to human health, necessitating the ongoing search for effective and sustainable therapeutic alternatives. This study underscores the potential of essential oils, specifically those derived from oregano, thyme, and clove, as potent antimicrobial agents. Through comprehensive GC–MS analysis, we identified high concentrations of active compounds—most notably eugenol in clove oil and carvacrol in oregano oil—correlating with their marked antimicrobial efficacy.

Our findings highlight the exceptional performance of these essential oils against a range of microorganisms, including multidrug-resistant strains. The study demonstrated that oregano and clove oils exhibited the lowest MIC values, affirming their superior antimicrobial properties. 

The synergy tests revealed that combinations of these oils generally produced additive or synergistic effects, enhancing their antimicrobial potency. Notably, a soap formulation incorporating clove and oregano oils proved to be as effective as synthetic antiseptics.

Given the growing challenge of antimicrobial resistance and the results found here, the integration of essential oils into antimicrobial formulations presents a viable solution. These natural compounds offer a promising, sustainable alternative to conventional synthetic antimicrobials, potentially mitigating public health threats. Further research and development are warranted to optimize these formulations and fully harness their therapeutic potential.

## Figures and Tables

**Figure 1 molecules-29-04682-f001:**
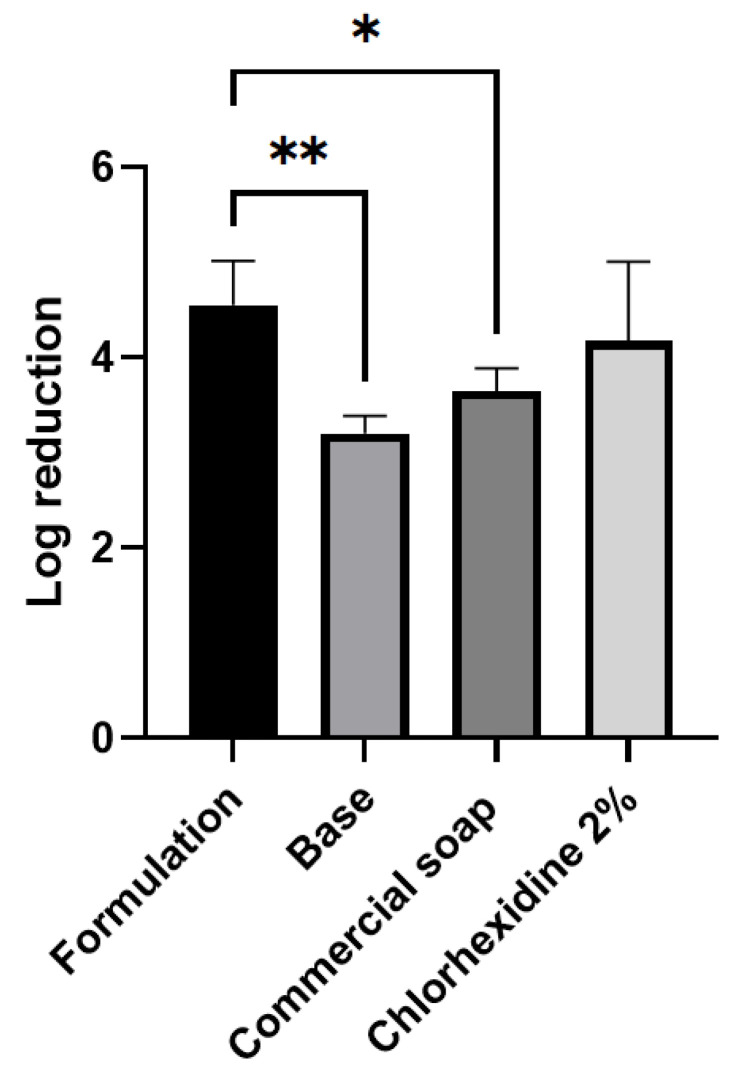
Comparison of in vivo antimicrobial activity of the different treatments. Log reduction indicates how much the contaminant level has been reduced as a result of the cleaning or sanitizing procedure (* *p* < 0.04, ** *p* < 0.002).

**Figure 2 molecules-29-04682-f002:**
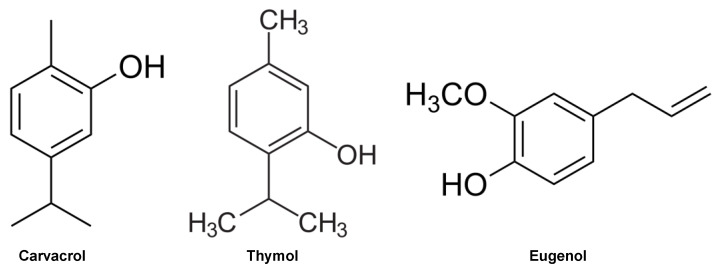
Molecular structures of the predominant compounds identified in the essential oils of oregano, thyme, and clove. This figure illustrates the key chemical constituents responsible for the characteristic properties and potential biological activities of these oils.

**Table 1 molecules-29-04682-t001:** Determination of the minimum inhibitory concentration (MIC) of essential oils for bacterial strains (in μL/mL for oils, µg/mL for gentamicin, and % for chlorhexidine).

	CLO	ORE	THY	GEN	CHLO
*E. coli* ATCC 25922	12.5	6.25	12.5	3.125	3.05 × 10^−5^
*E. coli* EW222	3.125	6.25	25	1.562	2.44 × 10^−4^
*E. coli* EW239	12.5	3.125	12.5	>50	3.05 × 10^−5^
*P. aeruginosa* ATCC 27853	25	25	50	6.25	1.95 × 10^−3^
*P. aeruginosa* S15	12,5	12.5	50	0.39	9.76 × 10^−4^
*S. aureus* ATCC 6538	25	6.25	12.5	6.25	1.22 × 10^−4^
*S. epidermidis* ATCC 12228	12.5	6.25	12.5	0.096	7.63 × 10^−6^
*S. marcescens* ATCC	1.56	3.125	6.25	0.78	1.9 × 10^−4^
*P. vulgaris* ATCC 6380	12.5	6.25	12.5	1.562	1.95 × 10^−3^
*S. Choleraesuis* ATCC 10708	6.25	3.125	6.25	12.5	9.76 × 10^−4^

Note: (CLO) clove; (ORE) oregano; (THY) thyme; (GEN) gentamicin; (CHLO) chlorhexidine.

**Table 2 molecules-29-04682-t002:** Determination of the minimum bactericidal concentration (MBC) of essential oils against tested bacteria.

	CLO	ORE	THY	GEN	CHLO
*E. coli* ATCC 25922	BC	BC	BC	BS	BS
*E. coli* EW222	BC	BC	BC	BS	BC
*E. coli* EW239	BC	BC	BC	*	BC
*P. aeruginosa* ATCC 27853	BC	BC	BC	BC	BS
*P. aeruginosa* S15	BS	BC	BC	BS	BS
*S. aureus* ATCC 6538	BC	BC	BC	BC	BS
*S. epidermidis* ATCC 12228	BS	BS	BS	BS	BS
*S. marcescens* ATCC	BC	BC	BC	BC	BC
*P. vulgaris* ATCC 6380	BC	BC	BC	BC	BS
*S. choleraesuis* ATCC 10708	BS	BS	BS	BC	BS

Note: (*) not inhibited; (BC) bactericidal; (BS) bacteriostatic; (CLO) clove; (ORE) oregano; (THY) thyme; (GEN) gentamicin; (CHLO) chlorhexidine.

**Table 3 molecules-29-04682-t003:** Determination of the minimum inhibitory concentration (MIC) of essential oils against filamentous fungi (in μL/mL for oils, μg/mL for fluconazole, and % for chlorhexidine).

	CLO	ORE	THY	FLU	CHLO
*A. fumigatus* ATCC 46645	3.125	0.003	0.195	250	<2.3 × 10^−5^
*A. fumigatus* LMC 9015.01	1.562	6.25	6.25	250	<2.3 × 10^−5^
*T. rubrum* ATCC MYA 3108	1.562	0.390	3.125	15.62	<2.3 × 10^−5^

Note: (#) inhibited all; (CLO) clove; (ORE) oregano; (THY) thyme; (FLU) fluconazole; (CHLO) chlorhexidine.

**Table 4 molecules-29-04682-t004:** Determination of the minimum inhibitory concentration (MIC) of essential oils against yeasts (in μL/mL for oils, μg/mL for fluconazole, and % for chlorhexidine).

	CLO	ORE	THY	FLU	CLHO
*C. neoformans* ATCC 90112	6.25	12.5	12.5	1.562	9.76 × 10^−4^
*C. albicans* ATCC 10231	12.5	3.125	12.5	7.81	2.44 × 10^−4^
*C. auris* CDC 811903	1.562	0.0976	0.781	3.9	9.76 × 10^−4^

Note: (CLO) clove; (ORE) oregano; (THY) thyme; (FLU) fluconazole; (CHLO) chlorhexidine.

**Table 5 molecules-29-04682-t005:** Determination of the minimum inhibitory concentration (MIC) of combined essential oils against tested bacteria (in μL/mL).

	CLO/ORE	CLO/THY	THY/ORE
*S. aureus* ATCC 6538	0.781	3.125	1.562
*S. epidermidis* ATCC 12228	3.125	6.25	3.125
*S. marcescens* ATCC	0.781	3.125	1.56
*E. coli* ATCC 25922	1.562	0.781	1.562
*E. coli* EW239	1.562	3.125	3.125

Note: (CLO) clove; (ORE) oregano; (THY) thyme.

**Table 6 molecules-29-04682-t006:** Fractional Inhibitory Concentration Index (ICIF) value and effect.

	CLO/ORE	CLO/THY	THY/ORE
*S. aureus* ATCC 6538	BS	BS	BS
*S. epidermidis* ATCC 12228	BC	BS	BS
*S. marcescens* ATCC	BC	BC	BC
*E. coli* ATCC 25922	BC	BS	BC
*E. coli* EW239	BC	BC	BC

Note: (BC) bactericidal; (BS) bacteriostatic; (CLO) clove; (ORE) oregano; (THY) thyme.

**Table 7 molecules-29-04682-t007:** Determination of the minimum bactericidal concentration (MBC) of combined essential oils against tested bacteria.

	CLO/ORE	CLO/THY	THY/ORE
*S. aureus* ATCC 6538	0.156 (Synergistic)	0.375 (Synergistic)	0.375 (Synergistic)
*S. epidermidis* ATCC 12228	0.75 (Additive)	1.00 (Additive)	0.75 (Additive)
*S. marcescens* ATCC	0.75 (Additive)	2.5 (Indifferent)	0.75 (Additive)
*E. coli* ATCC 25922	0.375 (Synergistic)	0.125 (Synergistic)	0.375 (Synergistic)
*E. coli EW239*	0.625 (Additive)	0.50 (Synergistic)	1.25 (Indifferent)

**Table 8 molecules-29-04682-t008:** Results of GC–MS analysis of clove, oregano, and thyme essential oils. The Kovats index (KI) is a system used to identify and characterize compounds based on their retention times in gas chromatography.

Identified Substance	KI	Clove (%)	Oregano (%)	Thyme (%)
tricyclene	926	-	-	2.42
p-cymene	1011	-	5.90	24.51
1.8-cineole	1020	-	1.80	1.49
trans-ocimeno	1048	-	2.41	4.49
terpinoleno	1085	-	1.95	5.25
camphor	1120	-	0.62	1.61
neoisotujol	1142	-	-	0.54
neotujol	1150	-	0.97	1.30
terpinen-4-ol	1162	-	0.54	0.91
thujanol	1174	-	0.57	-
thymol	1279	-	2.29	49
carvacrol	1287	-	72.63	2.83
eugenol	1336	84.36	-	-
trans-caryophyllene	1413	9.33	2.64	-
humulene	1446	1.46	-	-
caryophyllene oxide	1566	0.93	0.46	0.89

**Table 9 molecules-29-04682-t009:** Bacterial strains.

Microorganism	Reference	Origin
*Escherichia coli*	ATCC 25922	
*Escherichia coli*	EW222	public aquatic environments
*Escherichia coli*	EW239	public aquatic environments
*Pseudomonas aeruginosa*	ATCC 27853	
*Pseudomonas aeruginosa*	S15	soil
*Staphylococcus aureus*	ATCC 6538	
*Staphylococcus epidermidis*	ATCC 12228	
*Serratia marcescens*	ATCC 13880	
*Proteus vulgaris*	ATCC 6380	
*Salmonella choleraesuis*	ATCC 10708	

## Data Availability

The data are available on request from the corresponding author. https://www.mdpi.com/journal/molecules/instructions#suppmaterials.

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
