# Peer review of "Essential Oil-Based Soap with Clove and Oregano: A Promising Antifungal and Antibacterial Alternative against Multidrug-Resistant Microorganisms"

_molecules, 2024, doi:10.3390/molecules29194682_

Round 1

Reviewer 1 Report

Comments and Suggestions for Authors

This manuscript is describing about essential oil-based soap with clove and oregano for antifungal and antibacterial alternatives against multidrug-resistant microorganisms. The experiemts were relevantly designed and ineresting results were obtained. This manuscript should be revised at several points.

- Strain names appearing in the Abstract separately and Table should be presented with full spell.

- Microbial names should be italiclized.

- All numeral values should be presented with dot '.', not comma ','  for decimal point. In Table 2, superscript should be used for CHLO.

- In Tables 3 and 7, presentation should be shown using abbreviations. ex, Bacericidal -> BC; Bacteriostatic -> BS. Then, the abbreviations should be explained in the Table note. 

Comments on the Quality of English Language

Minor editing required.

Author Response

Response to Reviewer 1

2. Point-by-point response to Comments and Suggestions for Authors

Comments 1: Strain names appearing in the Abstract separately and Table should be presented with full spell.

Response 1: Strain names corrected

Comments 2: Microbial names should be italicized.

Response 2: All microbial names were reviewed and italicized.

Comments 3: All numeral values should be presented with dot '.', not comma ','  for decimal point. In Table 2, superscript should be used for CHLO.

Response 3: Numbers corrected

Comments 4: In Tables 3 and 7, presentation should be shown using abbreviations. ex, Bacericidal -> BC; Bacteriostatic -> BS. Then, the abbreviations should be explained in the Table note.

Response 4: Tables corrected

Reviewer 2 Report

Comments and Suggestions for Authors

It is very interesting work. But there are some obvious mistakes need to be corrected.

1. In Abstract and 2.2ï¼›Latin name of the microorganisms do not meet the format specification.

2. Page 4, line 141 “0,100-0,125” should be “0.100-0.125” ;

line 144 “0,125-0,155” should be “0.125-0.155” ?

3. Page 5, line 227; “0.150 nm to 0.460 nm” should be “0.150 to 0.460” ?

4. Page 7, line 300, “1.,56” should be “1.56”;

line 302, “3.,12” should be “3.12” ;

line 304, “12.,5” should be “12.5”.

5. Page 7, Table 2; The comma in the numbers should be decimal point?

The numbers in the last column are not expressed correctlly.

Some items in the note are not shown in the table, such as (X) not tested; (TEA)

tea tree; (MIN) mint; (EUC) eucalyptus; (ROS) rosemary;

6. Page 8, Table 3; Some items in the note are not shown in the table, please check.

7. Page 8, Table 4; Reference article 5 above.

8. Page 9, Table 5; Reference article 5 above.

9. Page 10, Table 9; What is “KI” mean.

10. Page 11, line 402-403; “0.,04” and “0.,002” should be “0.04” and “0.002” ?

Author Response

Response to Reviewer 2

2. Point-by-point response to Comments and Suggestions for Authors

Comments:

1. In Abstract and 2.2ï¼›Latin name of the microorganisms do not meet the format specification.

2. Page 4, line 141 “0,100-0,125” should be “0.100-0.125” ;

line 144 “0,125-0,155” should be “0.125-0.155” ?

3. Page 5, line 227; “0.150 nm to 0.460 nm” should be “0.150 to 0.460” ?

4. Page 7, line 300, “1.,56” should be “1.56”;

line 302, “3.,12” should be “3.12” ;

line 304, “12.,5” should be “12.5”.

5. Page 7, Table 2; The comma in the numbers should be decimal point?

The numbers in the last column are not expressed correctlly.

Some items in the note are not shown in the table, such as (X) not tested; (TEA)

tea tree; (MIN) mint; (EUC) eucalyptus; (ROS) rosemary;

6. Page 8, Table 3; Some items in the note are not shown in the table, please check.

7. Page 8, Table 4; Reference article 5 above.

8. Page 9, Table 5; Reference article 5 above.

9. Page 10, Table 9; What is “KI” mean.

10. Page 11, line 402-403; “0.,04” and “0.,002” should be “0.04” and “0.002” ?

Response 1: All points were reviewed and corrected

Reviewer 3 Report

Comments and Suggestions for Authors

The manuscript by Ana Paula Merino Cruz et al, entitled: Essential Oil-Based Soap with Clove and Oregano: A Promising antifungal and Antibacterial Alternative Against Multidrug-Resistant Microorganisms presents an interesting and useful study on the antimicrobial properties of Clove and Oregano oils.

The authors follow a structure that flows easy, is logical and is presented in a clear manner.

Nevertheless, there are some problems that need to be addressed:

-        The authors performed the antifungal and antibacterial activities only on the essential oils and not on the soap formulations.

-        The essential oils were dissolved in ethanol and its antimicrobial properties are well known. Did the authors not think to compare the results of the alcoholic solutions of the EOs with ethanol alone to measure the exact performance of the EOs?

-        How exactly was it decided to use the concentrations of the EOs in the final soap formulation? 50 µL/mL clove oil and 50 µL/mL oregano oil? Also, the ethanol was not used in the soap formulation.

Comments on the Quality of English Language

The English Language needs only minor revision

Author Response

Response to Reviewer 3

2. Point-by-point response to Comments and Suggestions for Authors

Comment 1: The authors performed the antifungal and antibacterial activities only on the essential oils and not on the soap formulations.

Response 1: Since the antimicrobial activity of the soap formulation is derived from the oils, the antifungal and antibacterial tests reflect the antimicrobial activity of the oils in synergism (see item 3.3: Antimicrobial Activity and Synergism of the Oils). Moreover, the evaluation of in vivo antimicrobial activity statistical analysis using the artificial contamination of S. marcescens (item 3.5) revealed a significant difference between the soap with essential oils and the base as well as compared to Commercial soap 1, thus confirming the superior performance of the formulation. Comparison with the base demonstrates that the difference in results occurred due to the addition of clove and oregano essential oils.

Comment 2: The essential oils were dissolved in ethanol and its antimicrobial properties are well known. Did the authors not think to compare the results of the alcoholic solutions of the EOs with ethanol alone to measure the exact performance of the EOs?

Response 2: After further dilution of the ethanol-oil mixture in the medium, ethanol exhibited no antimicrobial properties. Therefore, we chose not to present the ethanol data, although we added the following sentence to provide a clearer explanation.

“Ethanol was utilized as a solvent to dilute the oils, with its final concentration not exceeding 1% in the highest treatment. In all inhibition assays, ethanol was included as a control and demonstrated no inhibitory activity.

Reviewer 4 Report

Comments and Suggestions for Authors

The abstract should at least describe the methodology in one or two sentences.

The introduction contains all the necessary information.

It is not entirely clear to me why only oregano and clove oils were chosen for the soap and not thyme as well. This should be properly explained.

I also think that the y-axis signature in Figure 2 is quite unfortunate. It should be changed.

The discussion is correctly written. The literature used is appropriate and the authors have selected recent research that discuss the topic of interest.

Author Response

Comment 3: How exactly was it decided to use the concentrations of the EOs in the final soap formulation? 50 µL/mL clove oil and 50 µL/mL oregano oil? Also, the ethanol was not used in the soap formulation.

Response 3:  A sentence was added to the item 2.7 soap formulation

“This concentration is double the highest MIC value observed during the assays, which was 25 µL/mL against Pseudomonas aeruginosa ATCC 27853, for both clove oil and oregano oil. Thus, this ensures that the oils would inhibit the majority of strains.”

Response to Reviewer 4

2. Point-by-point response to Comments and Suggestions for Authors

Comment 1: The abstract should at least describe the methodology in one or two sentences.

Response 1: A sentence was added

“Using a range of in vitro and in vivo antimicrobial assays, including Minimal Inhibitory Concen-tration (MIC), Minimal Bactericidal Concentration (MBC), and Minimal Fungicidal Concentration (MFC), the essential oils were tested against a broad spectrum of pathogens. Additionally, the chemical composition of the oils was analyzed in detail using High-Performance Liquid Chromatography (HPLC).”

Comment 2: It is not entirely clear to me why only oregano and clove oils were chosen for the soap and not thyme as well. This should be properly explained.

Response 2: A sentence was added

“Due to the strong odor of thyme oil, it was deemed unsuitable for incorporation into the soap formulation.”

The aim was to use only a combination of two oils, which we believed would be sufficient to demonstrate enhanced potential.

Comment 3: I also think that the y-axis signature in Figure 2 is quite unfortunate. It should be changed.

Response 3: The correct term 'Log Reduction' was added, and its definition was provided in the legend.

Round 2

Reviewer 3 Report

Comments and Suggestions for Authors

Thank you for your replies